# Whole-Body Photobiomodulation Therapy for Fibromyalgia: A Feasibility Trial

**DOI:** 10.3390/bs13090717

**Published:** 2023-08-29

**Authors:** Bethany C. Fitzmaurice, Nicola R. Heneghan, Asius T. A. Rayen, Rebecca L. Grenfell, Andrew A. Soundy

**Affiliations:** 1Department of Pain Management, Sandwell and West Birmingham NHS Trust, Birmingham B71 4HJ, UK; arasu.rayen@nhs.net; 2School of Sport, Exercise and Rehabilitation Sciences, University of Birmingham, Birmingham B15 2TT, UK; n.heneghan@bham.ac.uk (N.R.H.); a.a.soundy@bham.ac.uk (A.A.S.); 3Clinical Research Facility, Sandwell and West Birmingham NHS Trust, Birmingham B71 4HJ, UK; rebecca.grenfell1@nhs.net

**Keywords:** fibromyalgia, chronic primary widespread pain, photobiomodulation therapy, whole-body, feasibility trial

## Abstract

Effective treatment for fibromyalgia (FM) is lacking and further treatment options are needed. Photobiomodulation therapy (PBMT) represents one potential treatment option. Whilst favourable findings have been reported using localised PBMT, no investigations have established the value of whole-body PBMT for the complete set of symptom domains in FM. A single-arm feasibility study was conducted in accordance with CONSORT (Consolidated Standards of Reporting Trials) guidelines. A non-probability sampling method was used to access individuals with FM. The primary outcome measure was identified as the Revised Fibromyalgia Impact Questionnaire (FIQR). Forty-nine participants were screened and twenty-one trial participants entered the trial. Nineteen participants completed the intervention (18 whole-body PBMT sessions over approximately six weeks). Descriptive statistics and qualitative analysis was undertaken to represent feasibility outcomes. Acceptability of the trial device and processes were established. Outcome measures towards efficacy data were guided by core and peripheral OMERACT (outcomes measures in rheumatological clinical trials) domains, utilising a combination of participant-reported and performance-based outcome measures. Data for the embedded qualitative component of the trial were captured by participant-reported experience measures and audio-recorded semi-structured interviews. Positive changes were observed for FM-specific quality of life, pain, tenderness, stiffness, fatigue, sleep disturbance, anxiety, depression and cognitive impairment. Patient global assessment revealed improvements at 6 weeks, with continued effect at 24 weeks. FM-specific quality of life at 24 weeks remained improved compared with baseline scores. The findings provided evidence to support a full-scale trial and showed promise regarding potential efficacy of this novel non-invasive treatment in an FM population.

## 1. Introduction

Fibromyalgia syndrome is a multisystem disorder characterised by a vast array of symptoms, including, principally, generalised body pain, fatigue, sleep and mood disturbance and impaired cognition [1]. Physical, emotional and cognitive functioning is significantly lower in FM patients compared with their age- and gender-matched counterparts [2,3]. It is the second most common rheumatological condition [4] and lifetime worldwide prevalence is 6.8–15% [5].

In addition to significant patient burden and direct medical costs, FM has a major socioeconomic impact with considerable indirect costs such as lost work productivity and disability benefits. A large US epidemiological study demonstrated annual healthcare costs to be three times higher in FM patients compared with age- and gender-matched controls ($9573 vs. $3291, respectively) [6]. A Canadian study revealed that less than half of FM patients were in employment, and of these, lost time from work due to FM symptoms was up to 4 weeks annually [7]. There is no cure for FM and long-term outcome data are limited—evidence shows chronicity to span at least seven years, and often much longer [8].

There is no known effective treatment for chronic primary widespread pain conditions [9] like FM, likely owing to their multifactorial aetiology and presentation [10]. It is commonplace for affected individuals to try a multitude of therapies, often accompanied with side effects despite the evidence of limited benefit [11,12]. The most recent National Institute for Health and Care Excellence (NICE) guidance [13] regarding chronic pain management advises against the use of many commonly instituted pain medications and interventions. The paucity of strong recommendations in international guidelines [14,15,16] highlights a need to explore other therapeutic methods and modalities. NICE called for further treatment options to be made available [17], identified PBMT as promising and recommended further research [16].

PBMT is a safe, non-invasive low-energy light (red and near infrared) therapy that is absorbed by endogenous chromophores to induce cellular changes [18,19,20]. Localised PBMT demonstrates positive results across a multitude of acute and chronic pain conditions [21,22,23,24,25,26,27,28,29,30,31,32,33,34,35]. National and international healthcare governing bodies recommend PBMT in treatment of cancer-related painful oral mucositis [19]. Conventionally delivered by a trained therapist using a small probe applied to specific painful areas, recent studies identified a need for larger probes and stipulated that novel delivery devices would be advantageous [24,35].

The development of whole-body devices has allowed participants to self-administer PBMT. The NovoTHOR^®^ device (THOR Photomedicine Ltd: Amersham, Buckinghamshire, United Kingdom) (Figure 1) delivers treatment to the whole body, requiring no specialist skills, and appears less labour-intensive and time-consuming [36]. Whole-body PBMT is a novel treatment modality with potential to address multiple aetiological mechanisms in patients experiencing chronic and diffuse pain. Co-existing features commonly include cognitive and emotional impairment and evidence is emerging that PBMT can aid in the treatment of these ailments [37].

The aim of this study was to investigate the feasibility of whole-body PBMT as a treatment option for reducing pain and pain-related co-morbidities in FM.

## 2. Materials and Methods

The following methods are laid out in accordance with the CONSORT extension to pilot and feasibility trials guidance [39] and SPIRIT-PRO Extension (SPIRIT, Standard Protocol Items: Recommendations for Interventional Trials; PRO, Patient-reported outcomes) guidance [40].

### 2.1. Trial Design

This was a single-centre and single-armed feasibility trial with an embedded qualitative component. All study procedures took place at Sandwell General Hospital’s Clinical Research Facility. Ethical approval was granted by the Health Research Authority (HRA) and Health and Care Research Wales (HCRW) (278452) and Leicester Central Research and Ethics Committee (21/EM/0231); ClinicalTrials.gov trial registration number NCT05069363; 10 June 2021. So that the intervention can be replicated when building on future research, the TIDieR (Template for Intervention Description and Replication) checklist was utilised [41]. The trial was designed according to the OMERACT hierarchy (Figure 2)—with the rationale that it clearly highlights a comprehensive view of the multidimensional nature of chronic pain, and subsequently provides the researcher with systematic and reproducible guidance.

The innermost circle of the OMERACT hierarchy contains the ‘core’ set of domains—the assessment of which are deemed to be essential in all FM clinical trials. The second concentric circle includes the outer core set of domains to be assessed in some but not all FM trials. The outermost circle includes the domains on the research agenda that may or may not be included in FM trials [42]. For the purposes of this study, any domain assessed that was not a ‘core’ domain was labelled ‘peripheral’ domain. We assessed all but two (CSF biomarkers and functional imaging) domains.

### 2.2. Participants

From January to June 2022, a non-probability sample was recruited from the Department of Pain Management at Sandwell and West Birmingham NHS Trust. Prospective participants were required to satisfy all inclusion criteria: widespread chronic pain of any origin (including axial pain, polyarthralgia, myofascial pain); able to provide informed written consent; ≥18 years; able to commit time to the trial treatment schedule of 6 weeks; score as low or moderate risk on the COVID-19 risk stratification tool—*applicable for the duration of the pandemic*. Exclusion criteria included: pregnancy; severe skin diseases (e.g., skin cancer, severe eczema, dermatitis or psoriasis); body weight ≥ 136 kg; uncontrolled co-morbidities (e.g., uncontrolled diabetes defined as HbA1c > 69 mmol/mol, decompensated heart failure, major psychiatric disturbance such as acute psychosis or suicidal ideation); use of systemic corticosteroid therapy including oral prednisolone or corticosteroid injections within the preceding 6 months; known active malignancy; inability to enter the NovoTHOR^®^ device or lie flat for 20 min (either due to physical reasons or other, e.g., claustrophobia); individuals speaking a language for which an interpreter cannot be sought (Oromo, Tigranian, Amharic, Greek). All participants gave written informed consent and were free to withdraw from study participation at any point.

### 2.3. Interventions

Screening of referred individuals was undertaken by the Principal Investigator (B.F.) via clinical records and a telephone call. A maximum of 5 mL of blood was taken as part of the screening process to confirm normal blood profile prior to trial commencement. Questionnaires were self-administered on paper in the presence of a study investigator (see Table 1). The study schedule is depicted in Figure 3 and Table 1, including an overview of events at each study visit.

The trial intervention is exhibited in Table 2 and NovoTHOR^®^ dosage parameters are shown in Table 3.

### 2.4. Outcomes

Eligibility criteria were explored by means of analysing eligibility rates. Refusal and retention rates were used to quantitatively assess acceptability. Qualitative interviews and participant-reported experience questionnaires were employed to evaluate the acceptability and practicability of the device, treatment schedule, trial design and appropriateness of outcome measures. OMERACT, established in 1992, is an international initiative to improve outcome measurement in rheumatology—affiliated with the International League for Rheumatology, World Health Organization and the Cochrane Collaboration Musculoskeletal Review Group [43]. FM is a good example of diffuse and widespread pain, encompassing both axial and multi-joint pain. The OMERACT hierarchy was used to assess treatment efficacy according to symptom domains. A combination of participant-reported (Table 4) and performance-based (Table 5) measures were employed. The following participant-reported outcome measures have all demonstrated reliability and validity in the assessment of pain conditions [44,45,46,47,48,49,50,51,52]. Additionally, these tools are recognised as the recommended standardised assessment tools for FM domains by an international consortium of experts in the field [53]. Table 1 depicts the time points at which questionnaires were administered. Table 4 gives a brief description of each tool used to assess outcome measures.

### 2.5. Sample Size

Sample size was estimated based on CONSORT guidelines for feasibility studies; describing a primary evaluation that focuses on descriptive analysis of feasibility/process outcomes (e.g., recruitment, adherence, treatment fidelity) [39]. Data from previous work surrounding localised PBMT in FM were used to inform sample size [71]. Our chosen sample size accounts for the study populations’ number of visits to pain clinics, study objectives and recommendations for the sample size calculations in pilot and feasibility trials [72,73,74,75]. Sample size for the qualitative component was guided by the concept of information power [74]. Considering past research [74] looking at experiences of an intervention, we attempted to interview all participants.

### 2.6. Statistical Methods

Feasibility data were assessed as the primary study outcomes. Descriptive statistics were utilised to report these data. Secondary outcomes to assess treatment efficacy comprise participant-reported and performance-based outcome measures. Microsoft Excel (2019) was employed to calculate mean averages and confidence intervals of parametric data. All results presented are mean average values with corresponding population standard deviations (±SD). Objective tenderness is depicted by scatterplots. An overview of pre- and post-treatment scores for all domains will be presented visually by means of box and whisker plots. Medication changes are depicted in tabular format. Skewness and Kurtosis and confidence interval analyses performed via IBM SPSS Statistics Versions 28.0.1.1 and 29.0.0.0 were employed to confirm normal distribution of the future primary outcome measure (FIQR) prior to implementing paired-samples T test using SPSS. Cohen’s *d* effect sizes were calculated using an online calculator [53] for all outcome measures. Cohen’s *d* effect size for the primary outcome measure were then employed to inform the sample size for the future definitive RCT [76]. Qualitative data gathered from semi-structured interviews will undergo reflexive thematic analysis [77].

## 3. Results

The data that support the findings of this study are available from the corresponding author, B.C.F., upon reasonable request.

### 3.1. Participant Flow

From January to June 2022, a total of 49 individuals were screened for potential enrolment onto the trial (Figure 4). Of these 49, 42 met the eligibility criteria and 24 gave consent and were prepped to commence the trial treatment. From January 2022, a total of 21 participants commenced treatment, with 19 completing the treatment schedule by August 2022.

### 3.2. Recruitment

Recruitment was via two sources: (i) self-referral via recruitment posters in pain clinics and procedure areas and (ii) clinician referral from pain clinics and pain intervention lists. The trial was advertised between January and August 2022. The first participant underwent their first treatment on 31 January 2022, with the final participant entering the trial on 29 June 2022. All participants had completed their treatment by 10 August 2022. Six-month data collection was completed 3 January 2023.

### 3.3. Baseline Data

All participants had clinician-diagnosed FM. Symptoms duration ranged from 4 to 31 years, with an age range of 28 to 66 years (14, 70% female; 6, 30% male). All but one male participant received 18 treatments. Further demographics and characteristics are shown in Table 6.

### 3.4. Primary (Feasibility) Outcomes

Quantitative data related to feasibility outcomes and guiding a definitive RCT are expressed below.

#### 3.4.1. Recruitment-Related Feasibility Outcomes

Eligibility: of the 25 participants that were excluded prior to consent, seven were excluded due to ineligibility; one became pregnant, two did not meet the inclusion criteria for pain type, two had received recent steroid injections and two had uncontrolled co-morbidities. Throughout the recruitment period, a considerable number of participants did not reach the screening phase due to having recently received steroids.Barriers to uptake: of the 18 participants that ‘declined to participate’, seven participants could not commit the time to the treatment schedule, three participants felt they would be too fatigued by the travel, one participant could not afford the petrol for the travel (lived more than 20 miles away), one participant was worried about personal unreliability due to unpredictability of flare ups, two participants were uncontactable, one participant had moved areas, one participant was actively trying to become pregnant and one participant was claustrophobic. The latter participant came to try the device but could not enter the study due to physical discomfort in the device and claustrophobia. One participant was commenced on a course of oral steroids during the latter stages of her treatment schedule in order to treat a respiratory infection.Trial retention: of the three participants that consented but did not proceed, one participant became pregnant, one became uncontactable and one participant re-considered due to both taxi costs and getting to top of the list for a steroid injection for their pain condition, and did not wish to postpone this. Subsequent to commencing the treatment schedule, one participant exited the trial after four ad hoc treatments due to difficulty with committing to the treatment schedule. The other participant exited the trial after completing 17 treatments secondary to a reported road traffic collision. All participants (*n* = 19) were contactable at a 6-month follow up.

#### 3.4.2. Trial-Related Feasibility Outcomes

Provision of information prior to trial: all participants were satisfied with the level of information they received prior to commencing the trial. One participant (5.3%) felt a video demonstration of the device prior to visiting the hospital might be helpful.Acceptability of treatment schedule: twelve participants (63.1%) were satisfied with the number and frequency of treatment sessions. One participant was ‘not sure’ and the remaining six participants (31.6%) said they would like to see a change in the number and frequency of treatment sessions. Of the latter six, five participants expressed a preference towards more frequent treatments (daily), longer treatment duration and an increased number of treatments over a longer time period. The remaining participant felt three days per week were too many visits. The same participant found the expense of transport an obstacle.Adherence to treatment schedule: for those participants who received the full treatment schedule, and the one participant receiving the majority of treatments, 50% participants (*n* = 10) received three treatments thrice weekly (Monday, Wednesday, Friday) for six weeks, as scheduled. Ten participants were non-adherent with the treatment schedule. These participants received all 18 treatments spanning a duration of 7–9 weeks, over which time 41 visits were postponed. Twenty-five visits (61%) were missed on the scheduled day attributable to medical reasons; ‘fibro flare’ (*n* = 2), fall (*n* = 1), poor sleep (*n* = 2), viral symptoms (*n* = 5), COVID-19 (*n* = 7), migraine (*n* = 1), burning sensations behind cheekbones (*n* = 4) and elective sinus surgery (*n* = 3). Practical reasons included lost car keys (*n* = 1), staffing and investigator availability (*n* = 3), dissatisfaction with travel expenses (*n* = 4), ‘Did Not Attend’ (*n* = 4) and ‘unforeseen circumstances’ (*n* = 1). Family reasons included a daughter having surgery (*n* = 1) and bereavement (*n* = 1). Work/study reasons included attending a course in Wales (*n* = 1).Acceptability of travel and expenses: single journey distance ranged from 0.6 miles to 9.6 miles (assuming the participant travelled from home). Participants’ distance travelled summated to a mean average ± SD of 181.45 miles ± 87.85 (range 22.8–364.8 miles). Thirteen participants (65%) travelled by car, two by bus, one by motorbike/scooter and one participant walked. Three participants travelled via taxi, one of which was through choice due to anxiety of driving and parking. Of the three that travelled by taxi, two reported difficulties relating to funding their journey. In one case, this led to missed appointments due to lack of funds, and the other participant who chose to come by taxi missed several appointments due to dissatisfaction relating to travel re-imbursement.Acceptability of participant-reported outcome measures: a total of 17 participants (89.5%) felt questionnaires administered were easy to follow and complete, with the remaining 10.5% (*n* = 2) being ‘not sure’. All participants (*n* = 19) felt the number and breadth of questionnaires was appropriate and necessary. Two participants (10.5%) felt more questionnaires were warranted to express further aspects of their condition impacting on their daily life. One participant felt that stiffness should have been measured objectively.Acceptability of performance-based outcome measures: a total of 17 participants (89.5%) found the Stroop Test delivered via mobile application straightforward to use and understood what was being asked of them. One participant (5.3%) was ‘not sure’, as they had no memory of performing the test. All participants (*n* = 19) reported that they would be happy to complete further additional cognitive objective measures in a future trial. All participants (*n* = 19) felt the tender point examinations were necessary towards assessing their condition and all would be happy for the same examination in a future trial. However, participants admitted they did not want considerable pressure applied at week 6 due to concerns over inducing a FM flare and no longer having the treatment available to aid this.Acceptability of audio-recorded semi-structured interviews: sixteen participants (84.2%) underwent audio-recorded semi-structured interviews. Fourteen of these participants found the interviews straightforward and felt comfortable. One participant did not answer and one participant felt a little uncomfortable due to not liking the sound of their voice.

#### 3.4.3. Treatment-Related Feasibility Outcomes

Acceptability of trial device: when asked to give comment about access and accessibility, six participants (31.6%) did not answer. The remaining 13 participants (68.4%) felt both the trial location and the device itself were easy to access. Constructive comments related to suggestion of a supporting rail for ease of entry and exit onto and off the device and a larger changing space. Two participants (10.5%) were asked to remove their transdermal fentanyl patch for every treatment. One participant managed to re-apply using adhesive dressings. The second participant required a temporary increase in quantity via prescription due to unsuccessful re-application of patches. All participants graded usability and comfort of trial device on a Likert scale from 1 = strongly agree through to 5 = strongly disagree (Figure 5).

2.Treatment satisfaction: participants were asked to list three words to describe their experience of the ‘light therapy pod’. Positive experiences included: helpful (*n* = 4), pleasant (*n* = 3), positive (*n* = 3), enjoyable (*n* = 2), comfortable (*n* = 1), efficient (*n* = 1), great (*n* = 1), useful (*n* = 1), interesting (*n* = 1), painless (*n* = 1), quick (*n* = 1), beneficial (*n* = 1), easy (*n* = 1), worthwhile (*n* = 1) and necessary (*n* = 1). One negative experience was described with regard to pain impeding ability to make appointments: difficult (*n* = 1). Low-energy positive emotions were: relaxing (*n* = 11), calming (*n* = 3) and soothing (*n* = 2). High-energy positive emotions were: pain relief (*n* = 4), warm (*n* = 3), better memory (*n* = 2), good mood (*n* = 2), better sleep (*n* = 1), more energy (*n* = 1), less confused (*n* = 1), reduced headaches (*n* = 1), clearer mind (*n* = 1), addictive (*n* = 1) and fun (*n* = 1). One future-related description was: hope (*n* = 1).3.Willingness towards future trial: all trial participants were willing to be involved in future research related to this device and all were happy with the prospect of a 50:50 chance of receiving 18 placebo treatments, selected at random, and being ‘blinded’ with goggles.

### 3.5. Secondary Outcomes

Standard deviations (SD) presented are with reference to sample SD. Figure 6 depicts a graphical representation of the median, mean, range and interquartile range for all outcome measures pre- and post-treatment. Confidence intervals and effect sizes for all outcome measures are demonstrated in Table 7 and Table 8. Additionally, pain-symptom-related medications and dosage changes post-treatment are reported.

#### 3.5.1. Core Domains: Participant-Reported Outcome Measures

Multidimensional function: pre-treatment FIQR scores were 79.7 ± 13.26. At week 6, scores had reduced to 55.3 ± 19.72—an improvement of 24.44 ± 20.38 points (*p ≤* 0.001). By week 24, scores were 65.68 ± 16.53; an increase compared with week 6 (*p* = 0.23), but clinically [78] and statistically significantly (*p* = 0.001) lower compared with baseline scores (Figure 7). FIQR score can be categorised by severity [78]. According to this scale, 17 participants (89.5%) commenced the trial with their FM symptoms having a severe effect on them and their symptoms being very intrusive. Six of the participants (37.5%) who commenced the trial with ‘severe’ FM (score ≥ 59 to 100), had only ‘moderate’ FM (score ≥ 39 to 59) after 6 weeks of PBMT, whilst four (25%) finished with treatment with ‘mild’ FM (score 0 to <39). Seven participants remained in the severe category, albeit all with a lower post-treatment score.

2.Pain: both pain-intensity and pain-interference scores showed clinically significant improvements post-intervention [78,79,80]. Pre-treatment pain-intensity was 7.08 ± 1.28. Post-treatment pain-intensity was 3.93 ± 1.38. Pain-interference score improved to 4.17 ± 1.99 from a pre-treatment score of 6.59 ± 1.32. A further question (which does not contribute to overall scoring) aims to ascertain the extent of relief from currently used analgesics—with improvements seen at week 6. Baseline perceived analgesic efficacy was 43.5% ± 17.55, rising to 53.89% ± 20.0 by week 6. All participants were confirmed to have FM, reflected in their scores of 25.1 ± 2.86 at baseline (comprised of WPI 15 ± 2.45 and SSS 10.1 ± 1.45). Scores improved to 16.21 ± 5.78 at week 6 (WPI 9.89 ± 4.21; SSS 6.32 ± 2.54). There is no reported MCID (minimal clinically important difference) for the 2016 Fibromyalgia Diagnostic Criteria, rather the American College of Rheumatology recommends use as a severity score in the longitudinal evaluation of participants [61]. When using the tool for its primary purpose—as a diagnostic tool—almost a third of participants (31.6%, n = 6) experienced an improvement in the order of magnitude that they would have been described as ‘negative’ for FM if were being assessed for diagnosis for the first time. At the commencement of each calendar week, each participant reported an average pain score out of 10 according to the NRS for pain for the preceding 7 days. There was a gradual decline in pain scores during the course of the trial. The average pain score reported at Visit 4 was 6.89. The average pain score reported at the start week 6 of treatment was 5.86.3.Fatigue: FSS pre-treatment score was 6.30 ± 0.86, reducing to 5.61 ± 1.16 post-treatment.4.Sleep disturbance: following six weeks of PBMT in this study sample, JSQ scores exhibited a reduction from additive score of 17.35 ± 1.90 (mean 4.34 ± 0.97) at baseline to 11.53 ± 6.17 (2.91 ± 1.74) post-treatment. The Jenkins Sleep Questionnaire (JSQ) categorises sleep into ‘little sleep disturbance’ and ‘high frequency of sleep disturbance’ [63]. All participants commenced the trial in the high frequency category, that is, difficulty falling to sleep and staying asleep, waking several times per night and feeling worn out after their usual night’s sleep. Ten participants (52.6%) fell into the ‘little sleep disturbance’ category post-intervention. Of those that demonstrated better sleep post-treatment (68.4%, n = 13), all improvements were ≥20% (range 20% to 88.9%), with an overall mean improvement of 33.6%.5.Patient global: post-treatment, participants were asked to rate the change to their overall quality of life, symptoms, emotions and activity limitation related to their pain condition. The mean average score was 5.47 ± 1.43. Four participants (21.1%) gave a score of 7. A further question denotes degree of change since commencing the treatment. At week 6, seventeen participants (89.5%) trended toward ‘much better’, whilst two participants scored ‘no change’. No participant trended towards ‘much worse’. By week 24, the mean average score was 3.79 ± 2.1, an indication that participants remained ‘a little better’ and ‘somewhat better’ at this timepoint. Thirteen participants (68.4%) continued to trend toward ‘much better’, five participants (26.3%) felt no change and one participant (5.3%) felt worse. Eleven participants (57.9%) had overall benefits in their condition in the order of ‘moderate’ or ‘substantial’ [79,81], with five participants (26.3%) reporting clinically significant improvements that were still ongoing at 24 weeks.

#### 3.5.2. Core Domains: Performance-Based Outcome Measures

Tenderness: the majority of participants did not tolerate the recommended pressure application of 4 kg/cm^2^ [54,68] across most tender points. The results are, therefore, presented according to the maximum pressure tolerated. Prior to commencement of the trial intervention, participants tolerated an average of 1.21 kg/cm^2^ ± 1.05 across each of the 18 recommended tender points. Post-treatment at week 6, participants tolerated higher pressures of 1.71 kg/cm^2^ ± 1.16. Average pain scores across 18 tender points (also known as Fibromyalgia Intensity Score or FIS) pre-treatment were 6.35 ± 1.84 compared with 5.17 ± 1.908 post-treatment. Figure 8 depicts the total MTPS score (sum of 18 NRS scores) for the corresponding total pressure tolerated when considering each tender point in isolation. A negative correlation can be seen post-treatment. That is, participants tolerated a higher pressure on examination for a corresponding lower pain score. It is clear that by the end of week 6, participants can tolerate a higher applied pressure for a corresponding lower MTPS score. Of the 342 total points examined pre-treatment, only three participants tolerated a pressure of 4 kg/cm^2^ across a collective of nine tender points. Post-treatment, five participants tolerated 4 kg/cm^2^ (27 points between them).

A notable change was demonstrated across 17 participants (94.4%). One participant refused to undergo tender point examination post-treatment. Summative total pressure tolerated pre-treatment ranged from 2.57 kg/cm^2^ to 62.04 kg/cm^2^, compared with range of tolerated total pressure post-treatment of 6.23 kg/cm^2^ to 67.25 kg/cm^2^. Post-treatment, three participants (16.7%) tolerated lower pressures, with the remaining fifteen demonstrating an improvement of 8% to 355% compared with their baseline measurements. Similarly, the ranges for MTPS scores were 72–172 pre-treatment and 37–135 post-treatment, demonstrating 9–69% reduction in pain scores in thirteen participants (72.2%). Of the remaining five participants with higher post-treatment pain scores, four tolerated a higher pressure during their examination. Figure 9 depicts the pressure change for corresponding MTPS score change, with the most optimal result being those participants represented in the top-left hand side of the graph, which shows that eleven participants (61.1%) tolerated higher pressures for corresponding lower pain scores post-treatment. Objective tenderness measures proved to be consistent with self-report measures compared with ‘fibrofog’ measures in this population (52.6% with complete consistency; 26.3% partial consistency).

#### 3.5.3. Peripheral Domains: Participant-Reported Outcome Measures

Anxiety and depression: depression scores post-treatment were 8.21 ± 3.68 compared to 12.5 ± 3.26 at baseline, representing a 34.3% reduction post-treatment. Similarly, anxiety scores exhibited a 24.8% reduction, which were 14 ± 3.71 pre-treatment and 10.53 ± 4.57 post-treatment. The HADS scale categorises anxiety and depression as mild, moderate and severe. A score ≤ 7 denotes non-cases [82]. All but one participant suffered with anxiety and depression at the outset of the trial (42.1% ‘severe’ anxiety; 26.3% ‘severe’ depression). Ten participants (52.6%) moved into a lower severity category of anxiety post-treatment, three of which improved by ≥2 categories. Five participants (26.3%) no longer suffered anxiety post-treatment and were classed as ‘non-cases’; one of which commenced the trial in the ‘severe’ category. Post-treatment, 78.9% of the participants (n = 15) moved into a milder category of depression than at the trial outset. Five participants (26.3%) improved by ≥2 categories, and 36.8% of the participants’ (n = 7) reported having their depressive symptoms resolved, being classed as ‘non-cases’ post-treatment.Stiffness scores pre-treatment were 9.05 ± 1.02, compared with 5.95 ± 2.56 post-treatment. Self-reported dyscognition also demonstrated improvement with a pre-treatment value of 8.35 ± 1.31, compared with 5.58 ± 2.56 post-treatment.

#### 3.5.4. Peripheral Domains: Performance-Based Outcome Measures

Dyscognition: the Stroop Test results are presented according to total correct score and accuracy (%). Total score achieved pre-treatment was 27.4 ± 16.0, compared with 31.21 ± 15.11 post-treatment. Accuracy was similar post-treatment (pre-treatment 85.23 ± 24.06; post-treatment 85.45 ± 24.04). When comparing self-report cognitive impairment to objective measures used, only four participants (21.1%) demonstrated absolute consistency, with a further five participants (26.3%) exhibiting relative consistency. Self-reported memory problems showed an overall mean improvement of 33.2% post-treatment.

### 3.6. Confidence Intervals and Effect Sizes

Of the participant-reported outcome measures, all but one demonstrated a large effect size—with the fatigue severity scale and tender point examination showing a medium effect size. The effect size for the Stroop Test was small. Confidence intervals aligned with this—no confidence interval crossed zero for all participant-reported outcome measures, but confidence intervals did cross zero for performance-based outcome measures. Prior to performing *t*-tests on the FIQR, skewness and kurtosis tests were undertaken, which confirmed the data to be in limits of a normal distribution [83]. The 95% confidence intervals and Cohen’s *d* effect sizes are summarised in for week 6 and week 24 in Table 7 and Table 8, respectively.

### 3.7. Medication Changes

Participants’ medications and dosages were compared before commencing the trial treatment and after completing the course of treatment (Table 9).

At trial outset, 14 participants were taking 17 opioid-based medications. Nine participants (64.3%) reduced or stopped opioid medication by week 6, including codeine phosphate, co-codamol 30/500, tramadol, oral morphine solution and morphine-based transdermal patches transdermal patches. A considerable number of antidepressants were being taken for the indication of depression (as opposed to neuropathic pain) and, as such, the dosing of these drug classes did not alter considerably. In particular, four participants were taking SNRIs (serotonin and norepinephrine reuptake inhibitors) in the form of duloxetine for the indication of neuropathic pain at trial outset. Two participants stopped duloxetine by the end of the trial, one of which commenced the trial on the maximum dose of 120 mg. Of those taking anticonvulsants for neuropathic pain, it is noteworthy that one participant’s baseline dose was double the maximum daily recommended dose. By week 6, the dose was within recommended limits. One participant was initiated on quetiapine by their psychiatrist for the indication of insomnia.

### 3.8. Power Calculation

One of the objectives following this pilot feasibility study was to inform the sample size of a definitive trial. The FIQR was taken to be the primary outcome measure for a RCT. Using Cohen’s *d* Effect Sizes, power tables for Effect Size d [76] were utilised to ascertain required sample size for a definitive trial. Using a two-tailed α = 0.01 and the Cohen’s *d* Effect Size for FIQR from the study population, 26 participants should be recruited into each arm for a 99% power. Aiming for 30 participants in both groups would allow for a 10% dropout rate in each group. This is a conservative estimate given the dropout rate in this pilot feasibility trial was <10%. Despite the large effect size obtained, 99% power has been suggested as a conservative estimate in view of the small sample size in this feasibility trial.

In comparison to similar trials looking at the same therapy (albeit a localised modality) in FM participants, a 2019 meta-analysis [84] reported a pooled standardised mean difference of 1.16. Out of the nine RCTs analysed, the range for sample size was between 10 to 25 in each arm. Therefore, based on previous average effect sizes, our proposed sample size is in keeping with past work, and again allows for a more conservative estimate.

### 3.9. Harms

Post-treatment physiological parameters did not reveal any adverse effects of treatment.

## 4. Discussion

### 4.1. Preliminary Feasibility Data Show Improvements in All OMERACT FM Domains Following a Course of Whole-Body Photobiomodulation Therapy

Evidence from systematic reviews and meta-analyses focusing on localised PBMT [84,85,86] has demonstrated consistent positive change across psychosocial domains for people with FM. Moreover, no side effects were reported. For the PGIC scale, clinically significant improvements were reported as ‘much improved’ or ‘very much improved’ [87,88,89], equivalent to a score or 6 or 7 [65]. In the present study, participants’ quality of life improved significantly, both statistically and clinically—with this effect continuing at 6 months [78]. Studies have rarely followed up fibromyalgia patients receiving photobiomodulation therapy for as long as 6 months [84]. In general, in pain management communities, it is expected that the positive effects of physical treatments do wane with time [90], even in more discrete, image-diagnosed conditions that encompass locally targeted treatments [91]. Latest FM guidelines highlight the inconsistent nature of long-term follow-up across a range of pharmacological and physical treatment entities [92]. There is no one explanation for this, and as with the nature of pain itself, it is usually multi-factorial and encompasses the whole spectrum of biopsychosocial factors [93]. It is, therefore, not surprising to see a rise in FIQR scores at 6 months in our population. This factor needs to be quantified and compared against controls in future trials.

Pain reduction was clinically significant [81,94] with significant improvement in pain-intensity and pain-interference ‘categories’ [81,94,95]. This demonstrates consistency with the aforementioned FM trials [84,85,86] and was also supported by the reduction in pain-related medication reported in this study. The current study identified participants having an increased pressure pain threshold across widespread muscle groups following a course of whole-body PBMT—which is consistent with data from a recent whole-body PBMT RCT [96].

Self-reported stiffness demonstrated a clinically significant improvement more than double the MCID of 13% for FIQ stiffness [79]. A significant improvement in stiffness was observed in localised PBMT meta-analyses in FM patients [84]. This study population had high baseline fatigue levels, which improved post-treatment but remained in the severe category [97]. The change was twice as much as the MCID [89,98] and was consistent with meta-analysis data regarding localised PBMT [84]. Fatigue is multidimensional and its aetiological mechanism is not fully understood [98]. Several recent meta-analyses evaluating localised PBMT in healthy participants have identified improved fatigue-related outcomes [99,100,101,102]. Further research is needed around this for people with FM. With regard to sleep, a recent Cochrane review suggested a JSQ MCID value of 20% [103]—a value which 100% of the study population achieved. Localised PBMT meta-analyses have not explicitly reported on sleep [84].

The current study illustrates improvements in anxiety and depression more than twice the MCID—taken as a change of 1.7 based on other chronic diseases [104]. The aforementioned meta-analyses mirror these improvements [84]. Moreover, recent systematic reviews and meta-analyses have demonstrated significant reductions in anxiety, depression, traumatic brain injury with co-morbid depression and post-traumatic stress disorder [105,106,107]. Promising meta-analytic data from using localised PBMT for dementia treatment [108] and for healthy adults [109] is worth noting. Specifically, executive function, memory and selective inattention improved, which, interestingly, are the subdomains with most substantial deficit exhibited in FM patients [45,69].

### 4.2. Proposed Biochemical Mechanisms of Photobiomodulation Therapy and Its Relation to FM Pathogenesis

From a biochemical viewpoint, mitochondrial dysfunction with a resultant increase in reactive oxygen species (ROS) underpins FM pathogenesis [110,111,112]. There is reduced oxygen supply to muscles secondary to a reduction in capillary quantity and function [112]. Suggested therapeutic strategies should, therefore, specifically target reducing oxidative stress [110,113]. PBM light (red and near infrared) comprise optimal wavelengths to achieve both tissue penetration and absorption at a mitochondrial level, ultimately reducing oxidative stress and increasing adenosine triphosphate (ATP) synthesis. Downstream effects are chemotactic, neural, lymphatic and humoral—culminating in improved blood flow and oxygen delivery, reduced oedema and improved tissue repair, anti-inflammatory and analgesic effects [114]. In healthy participants, meta-analyses showed reduced lactate levels and other biomarkers of post-exercise muscle damage following both localised and whole-body PBMT [101,102,115,116]. The potential to improve muscle functioning and recovery may explain why important factors such as pain interference and ability to cope has significantly improved in our population.

### 4.3. Whole-Body Treatment Approach Could Be Advantageous in the FM Population

Emerging evidence describes a small-fibre neuropathy associated with FM pain [117]. Systematic reviews and meta-analyses showed localised PBMT to be effective in treating neuropathic pain [110]. FM represents a more widespread and resistant form [117], thus lending itself to the whole-body approach. Furthermore, meta-analysis data regarding localised PBMT shows improvements in localised muscle pain [118]. There appears to be a clear advantage in supporting use of whole-body PBMT in the FM population.

A study evaluating whole-body PBMT in a group of female athletes resulted in better quality sleep and, therefore, a reduction in quantity of sleep, even after higher training loads [119]. Augmented autonomic profiles seen after whole-body PBMT [96,119] has implications on sleep quality and vice versa [120,121]. The improvements seen in our population with regard to sleep, psychological symptoms and cognition may be explained by the superadded central stimulation seen with whole-body PBMT, whereby sufficient brain penetration is achieved to improve sleep and other psychological factors [96,106,107,119].

### 4.4. Limitations

Several limitations are acknowledged from the current results. Standard design and bias limitations associated with feasibility studies are accepted. Specifically, a small sample size was selected in order to assess the primary aim of feasibility of trial device and procedures. Therefore, the sample size was not powered as for a full trial due to the nature of this pilot study, and was indeed deemed premature to perform a multi-centre controlled trial prior to obtaining feasibility date. For the aforementioned reasons, placebo therapy was not included this trial and, as such, we cannot rule out the placebo effect, at least in part, contributing to our positive results. Safety data were based on clinical symptoms and physiological variables. Trial duration spanned four seasons; hence, weather as a variable could not be standardised and weather can affect FM severity [122,123,124,125]. Pain catastrophizing was not controlled for; however, it is a factor that maintains chronic pain and predicts a poorer response to treatment [126,127]. Our data may not be representative of the FM population in general. The reason for this is past population studies [55,128] identified a lower baseline score. This suggests our study could have included participants with more severe cases of FM. A future national multicentre trial will give a better understanding of baseline severity in those participants being treated under a secondary care pain clinic.

### 4.5. Recommendations for Future Research and Clinical Implications

With regard to MTPS, it should be noted that results are subject to how a participant is feeling on a given day [129]. FM is regarded as an unpredictable condition with significant temporal variability in symptoms [130] and this is reflected in the scores. However, the MTPS remains the recommended method for assessing tenderness [54]. Given the current results, an update to the recommended applied pressure needs could be considered to a lower value.

Objective assessment is an alternate method of stiffness analysis, but none have been recommended specifically for FM. Quantification of taut muscle bands in stiffness assessment is one possibility [131]; however, there is no consensus regarding the standardisation of muscle groups to be assessed. Furthermore, studies did not reliably show that quantitative stiffness correlates to symptom burden in the FM population [132,133]. Recent studies have shown promise in this technique in assessing post-stroke stiffness [134], but further research to validate this technique in the FM population is warranted.

There is no standard recommended objective cognitive test for FM patients [45,54]. Our results showed improved processing speed post-treatment, but only with a small effect size. Stroop accuracy (surrogate marker of inhibitory control) was unchanged. Assessment is further confounded by heterogeneity in mood levels which are known to impact cognition [98].

We recommend the FIQR as the primary outcome measure for a future definitive trial. The original FIQ is the primary outcome measure in many FM trials [84]. It is specific to FM and it is simple to interpret—giving one overall figure which encompasses function, overall impact and symptoms of FM. Moreover, FIQR correlates directly with the original FIQ [55], which means that it is comparable not only with present and future scores but also with historical FIQ scores. Future research will be guided by the updated Medical Research Council and National Institute for Health Research [135].

Recruitment was negatively impacted by the number of participants receiving steroid injections. Animal studies have shown reduced anti-inflammatory effects of PBM in those using steroids concomitantly [136]. However, data regarding the degree of hypothalamic-pituitary-adrenal (HPA) axis suppression are inconsistent. For instance, evidence [137] has identified that following steroid injections normal function in the HPA axis returns by day 28. This has implications for eligibility criteria and would increase recruitment for a subsequent trial.

From a clinical viewpoint, we cannot ignore the wealth of recent clinical guidelines from national and international governing bodies recommending the use of PBMT for various painful conditions: the World Health Organization [29], International Association for the Study of Pain [32], NICE [19], American College of Physicians [28] and British Medical Journal [21]. NICE acknowledges the short-term clinical utility and safety of localised PBMT for chronic primary pain and recommends the assessment of effectiveness of PBMT on sleep, pain interference and physical function, long-term effectiveness and assessment of cost-effectiveness [13]. The present study demonstrated improvements in all but cost-effectiveness, which was not assessed. Cost-effectiveness so far is promising and has been demonstrated for the use of PBMT in oral mucositis and myofascial temporomandibular disorders [138,139,140]. The definitive trial should consider incorporating participants receiving steroid therapy and undergoing separate sub-group analyses. Participants should potentially be presented with a choice of two treatment schedules to introduce a degree of flexibility.

## 5. Conclusions

The findings indicate high acceptability of trial device and procedures amongst the FM population. FM remains a challenge, and this pioneering work has shown significant improvements in all fibromyalgia domains in the short and longer term. In a condition renowned for its shortfall in efficacious treatments, we now owe it to our patients to pursue the real-world approach towards widely instituting this safe, non-invasive treatment. The remaining questions to be answered are whether this is truly more effective and cost-effective than usual care.

## Figures and Tables

**Figure 1 behavsci-13-00717-f001:**
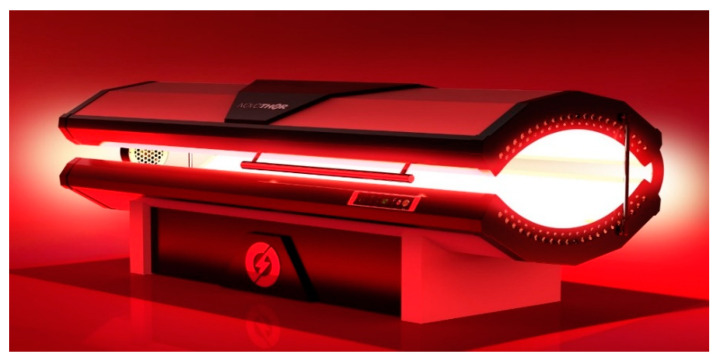
NovoTHOR^®^ whole-body PBMT device. Reprinted with permission from THOR Photomedicine Ltd. [38], 2019–2023, THOR Photomedicine, novothor.com (accessed on 8 December 2021).

**Figure 2 behavsci-13-00717-f002:**
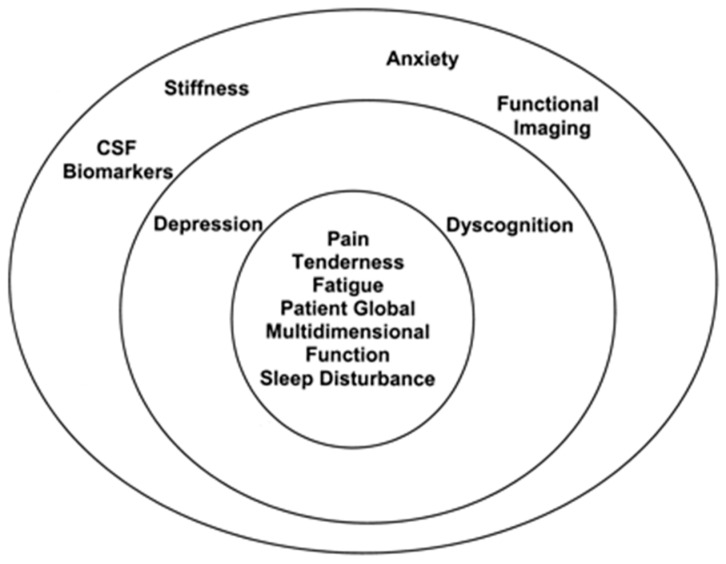
OMERACT hierarchy of domains. Reproduced with permission from OMERACT Fibromyalgia Working Group [42], 2023, OMERACT, www.omeract.org (accessed on 26 January 2021).

**Figure 3 behavsci-13-00717-f003:**
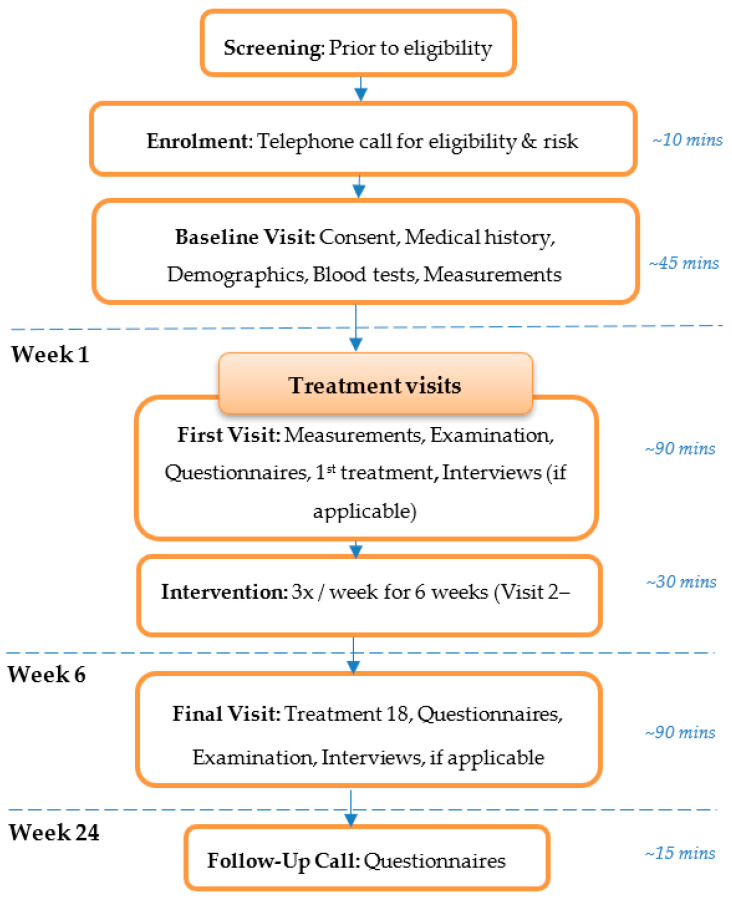
CONSORT study flow diagram.

**Figure 4 behavsci-13-00717-f004:**
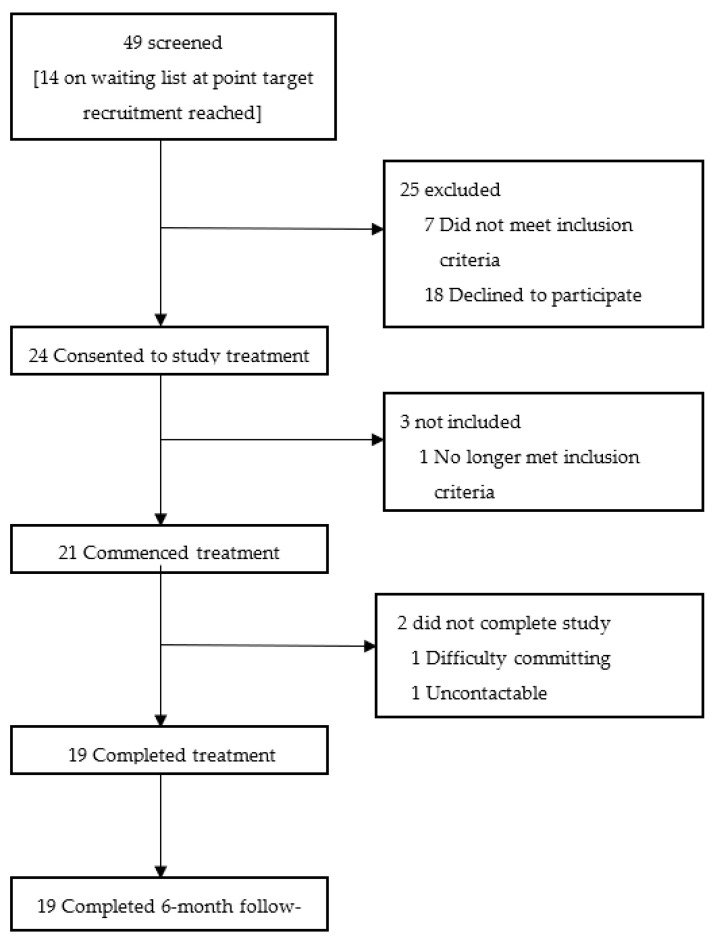
CONSORT Flow Diagram.

**Figure 5 behavsci-13-00717-f005:**
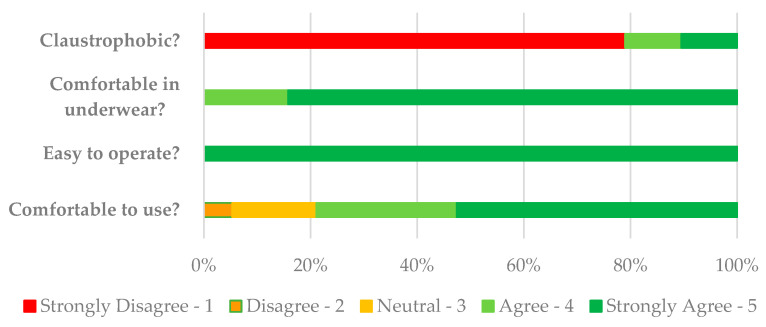
Participant experience of trial device.

**Figure 6 behavsci-13-00717-f006:**
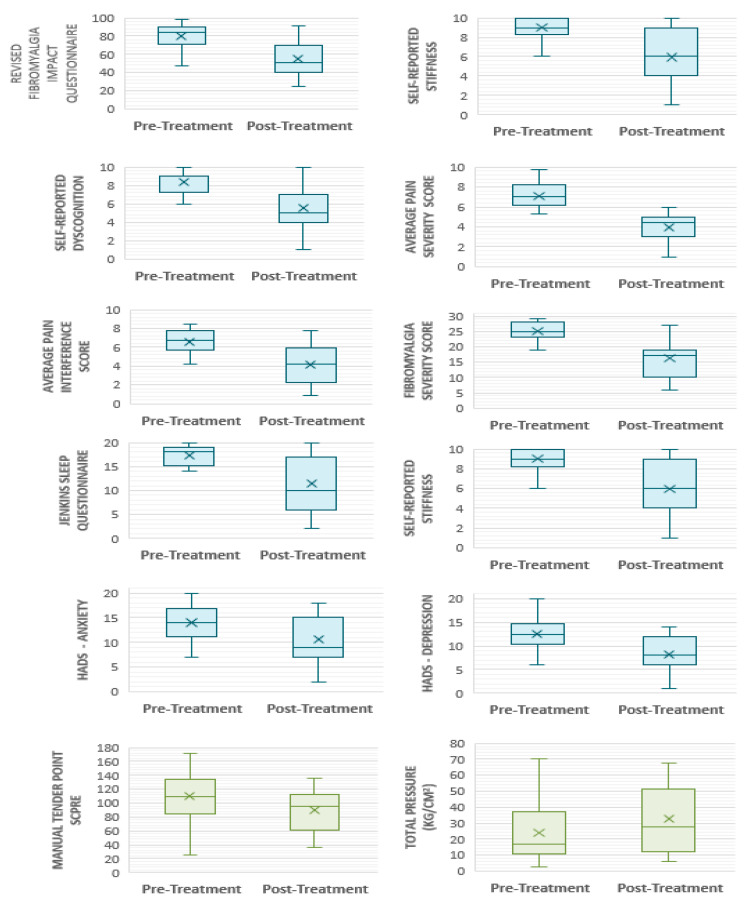
Box and whisker plots demonstrating improvements following trial intervention for all outcome measures. Blue denotes participant-reported outcome measures. Green denotes performance-based outcome measures.

**Figure 7 behavsci-13-00717-f007:**
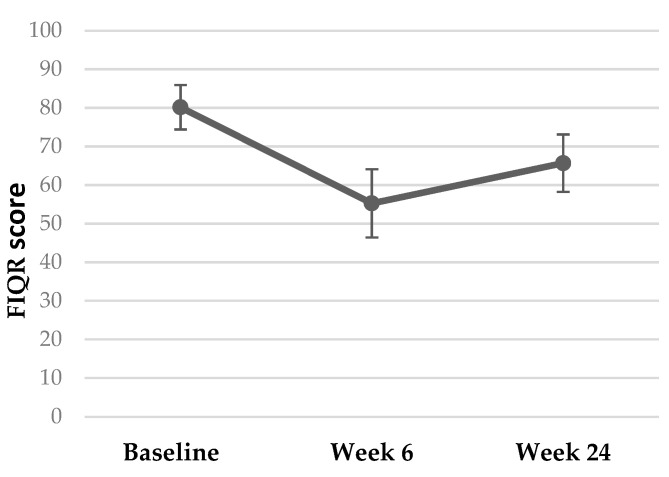
Mean FIQR scores (*y*-axis) with 95% Confidence Intervals, at specified timepoints (*x*-axis).

**Figure 8 behavsci-13-00717-f008:**
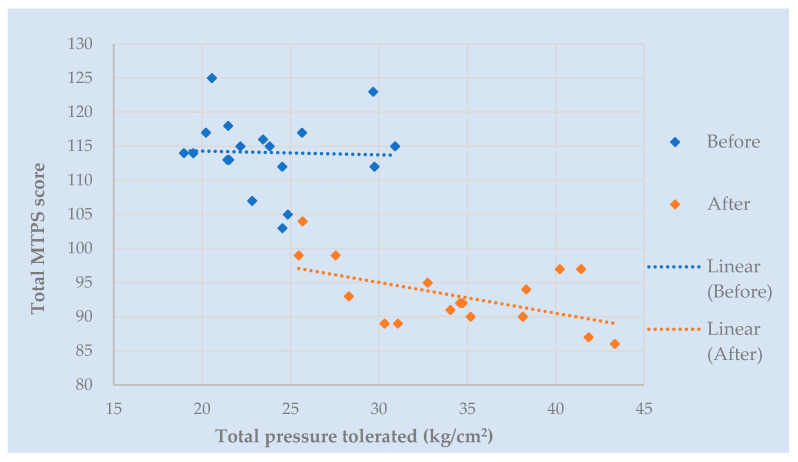
Manual Tender Point Survey score (*y*-axis) versus pressure tolerated (*x*-axis), representing an average score for tender point across anatomical location. Each diamond represents one of the 18 anatomical locations. *y*-axis demonstrates total sum of MTPS (Manual Tender Point Survey) scores for each tender point. *x*-axis shows total pressure tolerated for each tender point. Blue diamonds represent scores for each tender point pre-treatment. Red diamonds represent scores post-treatment. Dotted lines depict correlations.

**Figure 9 behavsci-13-00717-f009:**
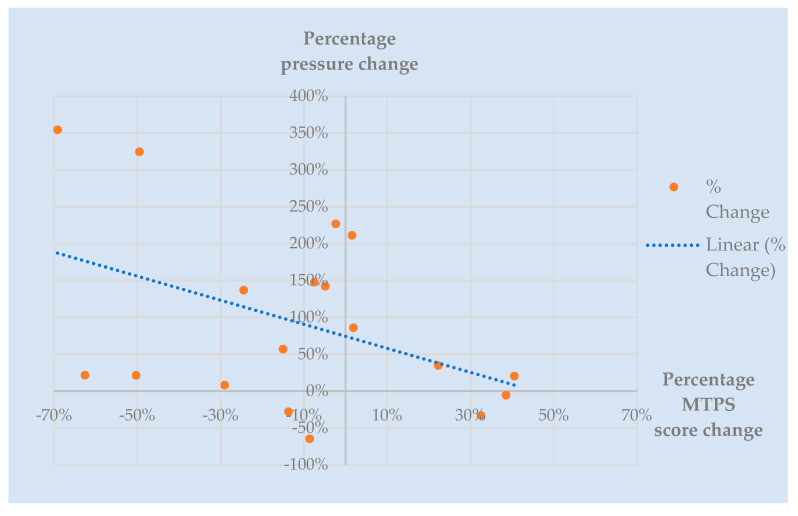
Percentage change in pressure and pain scores for each participant. *y*-axis describes percentage change in total pressure tolerated post-treatment at week 6. *x*-axis describes percentage change in MTPS (Manual Tender Point Survey) scores post-treatment. Each dot denotes an individual participant.

**Table 1 behavsci-13-00717-t001:** Outline of study procedures.

Procedures	Telephone Call	Baseline Visit	First Visit	Visit 2–Visit 17	Final Visit	6-Month Follow-Up
Eligibility Assessment	✓					
Informed consent		✓				
Blood Tests: Full blood count, Urea and electrolytes, Liver function tests, HbA1c (if diabetic)						
Demographics: Age, Gender, Marital status, Employment status, Educational level, Ethnicity		✓				
Medical History: Chronic pain symptom duration, Co-morbidities, Medications		✓				✓
Measurements: Height, Weight, BMI, Blood pressure, Heart rate, Oxygen saturations		✓			✓ ✓	
Participant-reported outcomes measures (PROMs): Brief Pain Inventory Widespread Pain Index/Symptom Severity Score Fatigue Severity Scale Jenkins Sleep Questionnaire Patient Global Impression of Change Revised Fibromyalgia Impact Questionnaire Hospital Anxiety and Depression Scale		✓ ✓ ✓ ✓ ✓ ✓			✓ ✓ ✓ ✓ ✓ ✓ ✓	✓ ✓
Performance-based outcome measures (PBOMs): Tender Point Count Stroop Test		✓ ✓			✓ ✓	
Treatment			✓ ✓	✓	✓	
Weekly Numerical Rating Scale (NRS) Participant-reported experience measure (PREM)				✓	✓ ✓ ✓	
Audio-recorded qualitative interviews (optional)			✓	✓	✓ ✓	

**Table 2 behavsci-13-00717-t002:** Template for intervention description and replication (TIDieR) checklist.

Brief Name	➢ Whole-Body Photobiomodulation Therapy—18 sessions
Why	➢ Eighteen sessions are the currently recommended and widely instituted and accepted practice with the NovoTHOR^®^ device. ➢ This device was developed in 2013, and since then, 251 NovoTHOR^®^ systems have been developed of which 217 systems are still in regular use, treating at least four patients per device per day. This equates to approximately 1.6 million treatments since its inception. No significant adverse events have been reported to date.
What	➢ All participants entering the trial will receive a course of whole-body PBMT. ➢ The NovoTHOR^®^ Whole-Body PBMT device consists of a hinged, clamshell design with light-emitting diodes (LEDs) arranged to emit near-infrared and visible red light → PBMT is delivered to the entire body at once. ➢ A Participant Information Sheet (PIS) will be provided at least 48 h before participants are requested to consent to the study. They will be given the opportunity to undertake an experience session. ➢ Participants will be expected to lie horizontally in the device with the lid as closed as they are comfortable with.
Who provided	➢ All trial investigators, following a short training session in the use of NovoTHOR^®^.
How	➢ The LED equipment delivers red and near infrared light therapy to the participant (as per the settings illustrated in Table 2).
Where	➢ Clinical Research Facility, SWB NHS Trust. ➢ Participants are registered at the Trust and are, therefore, geographically within the region. ➢ The device requires a well-ventilated, spacious, temperature-controlled room, with appropriate mains electricity.
When and how much	➢ Session 1 = 6 min. ➢ Session 2 = 12 min. ➢ Sessions 3–18 = 20 min. ➢ Timescale: 3 treatments/week for 6 weeks. ➢ The dosage of LED light (also known as ‘fluence’) will be equivalent to 25 J/cm^2^. The device will supply a dual wavelength of red and near-infrared light with a 50:50 ratio; 660 nm and 850 nm, respectively.
Tailoring	➢ After liaison with experienced clinicians within the field with experience dealing with our population in the NovoTHOR^®^, we decided to slowly uptitrate the treatment times during the first three treatments for all participants.
Modifications	➢ Described in ‘Results’ section.
How well	➢ Described in ‘Results’ section.

**Table 3 behavsci-13-00717-t003:** NovoTHOR^®^ Parameters.

NovoTHOR^®^ Parameters		Unit
Wavelengths of red and near-infrared (NIR) LEDs 50:50 ratio	660 850	nm nm
Number of LEDs	2400	
Power emitted per LED	0.289	W
Beam area per LED (at the lens/skin contact surface)	12.0	cm^2^
Total Power emitted	694	W
Total Area of NovoTHOR^®^ emitting surfaces	26,740	cm^2^
Treatment Time	1200	s
Continuous Wave (CW) (not pulsed)	CW	
Irradiance	0.028	W/cm^2^
Fluence	33.6	J/cm^2^

**Table 4 behavsci-13-00717-t004:** Participant-reported outcome measures (PROMs).

OMERACT Domain and Outcome Measures	Tool Background, Use and Scoring
‘Core’ Domains:	
Multidimensional function FIQR ^(*i*)^ (2009, replacing FIQ)	Recommended outcome measure in assessment of ‘multidimensional function’ or health-related quality of life [54]. A total of 21 questions across 3 domains: ‘function’, ‘overall impact’, ‘symptoms’. Each question requires a score based on an 11-point NRS pertaining to previous seven days, with a score of 0 being the ‘best’ and 10 being ‘worst’. Administrator calculates overall score. Nine questions from Domain 1 are totalled and divided by 3. Two questions from Domain 2 are simply added. Ten questions from Domain 3 are totalled and divided by 2. The final sum of resulting 3 figures represents the total (0 to 100). Higher scores indicate increased severity of FM [55].
Pain BPI-SF ^(*ii*)^ (1994)	Distinguishes pain into two components in preceding 24 h—pain intensity and pain interference [56]. The recommended pain assessment tool in FM clinical trials [54]. ‘Sensory dimension’: asked to rate ‘worst’, ‘least’, ‘average’, ‘pain now’ on 11-point NRS. ‘Reactive dimension’: score extent pain has interfered with mood, walking and other physical activity, work, social activity, relations with others, and sleep (0 = ‘does not interfere’, 10 = ‘completely interferes’) [56]. Four pain-intensity and seven pain-interference results averaged to give overall pain-intensity score and pain-interference score (0 to 10), respectively [57,58,59].
WPI + SSS ^(*iii*)^ (2010, updated 2016)	Updated diagnostic tool and a potential alternative [59] to original tender point examination, 1990 [60]. WPI: tick painful anatomical areas in preceding week. Nineteen areas are listed across 5 anatomical regions; 4 of which need to be ‘positive’ for an initial diagnosis of FM to be met. SSS: scored out of maximum of 12. Encompasses array of symptoms—user asked to report their presence and/or severity. Total potential combined WPI-SSS score is 31—higher scores indicate more severe FM [59]. Updated 2016 version: for user to be positive for FM diagnosis must score WPI ≥ 7 and SSS ≥ 5, or WPI 4–6 and SSS ≥ 9 [61].
Fatigue FSS ^(*iv*)^ (1989)	Unidimensional generic fatigue rating scale [62], emphasises functional impact of fatigue [63]. The recommended fatigue assessment tool for FM [54]. Nine fatigue-related questions, each scored on a 7-point Likert agreement scale (1 to 7). Resultant score is average of 9 scores, with maximum possible score of 7—indicating the most severe fatigue-related symptoms and intrusiveness.
Sleep disturbance JSQ ^(*v*)^ (1988)	Four-item self-report questionnaire designed to measure frequency of sleep problems in past month. The recommended assessment tool to evaluate sleep in FM patients [54]. A 5-point Likert scale (0 = ‘not at all’ to 5 = ‘22–31 days’) was utilised to evaluate the number of days/month that specific sleep-related issues occur (trouble falling and staying asleep, waking up several times/night, waking up after usual amount of sleep feeling tired and worn out). Maximum possible score is 20. Higher scores indicate higher frequency of sleep problems [64].
Patient Global PGIC ^(*vi*)^ (1970s)	Self-report global change questionnaire: 7-point NRS (1 to 7) to determine degree of change following a treatment from patients’ own perspective. Score of ‘1’ indicates either no change or worsening symptoms since treatment. A ‘7’ indicates the patient feels ‘great deal better, considerable improvement that has made all the difference’ [65]. IMMPACT (Initiative on Methods, Measurement and Pain Assessment in Clinical Trials) recommended for evaluating participant ratings of overall improvement in pain treatment trials [65]. Specifically recommended in the assessment of global improvement of FM patients in conjunction with the FIQR [54].
‘Peripheral’ Domains:	
Anxiety HADS ^(*vii*)^ (1983) *Anxiety subsection* *(HADS-A)*	A 14-item measure: each item rated on a 4-point severity scale (0 to 3). HADS-A subscales: comprised of 7 items. Acknowledged to have been used in FM trials assessing medication efficacy [54].
Depression HADS ^(*vii*)^ *Depression subsection* *(HADS-D)*	HADS-D subscales: comprised of 7 items. Scores range from 0 to 21. Higher scores indicate more severe symptoms [66,67]. The recommended tool for assessment of depressive symptoms in FM patients [54].
Stiffness *Subsection of FIQR* ^(*i*)^	
Dyscognition *Subsection of FIQR* ^(*i*)^	

(*i*) FIQR (Revision Fibromyalgia Impact Questionnaire), (*ii*) BPI-SF (Brief Pain Inventory—Short Form), (*iii*) WPI-SSS (Widespread Pain Index—Symptom Severity Score), (*iv*) FSS (Fatigue Severity Scale), (*v*) JSQ (Jenkins Sleep Questionnaire), (*vi*) PGIC (Patient Global Impression of Change), (*vii*) HADS (Hospital Anxiety and Depression Score).

**Table 5 behavsci-13-00717-t005:** Performance-based outcome measures (PBOMs).

OMERACT Domain and Outcome Measures	Tool Background, Use and Scoring
‘Core’ Domains:	
Tenderness Tender Point Examination (1990)	Manual Tender Point Survey/Fibromyalgia Intensity Score (MTPS/FIS) method is validated for FM population [68]. The currently recommended tenderness assessment for FM trials [54]. Eighteen specific tender points (9 bilateral anatomical areas) identified by American College of Rheumatology in 1990 [60]. Assessed with hand-held Wagner FORCE TEN^TM^ FDX pressure algometer—incremental increase up to a maximum of 4 kg/cm^2^. Pain severity rated at each point according to verbal NRS, with NRS ≥ 2 ‘positive’ for a tender point. Anatomical points: low cervical (C5-C7); 2nd rib (2nd costochondral junction); greater trochanter (posterior to trochanteric prominence); knee (at medial fat pad proximal to joint line); occiput (at suboccipital muscle insertions); trapezius (a midpoint of upper border), supraspinatus (above scapular spine near medial border), lateral epicondyle (2 cm distal to epicondyles); gluteal (upper outer quadrants of buttocks in anterior fold of muscle).
‘Peripheral’ Domains:	
Dyscognition Stroop Test (1935, original)	Selected in attempt to address the cognitive domains of inhibitory control, processing speed and memory, which have been shown to be the most significant cognitive complaints in the FM population [45,69]. The Stroop Test for Research application [70] is a computer-based test, performed via mobile application in the current study. A series of colours are spelt out on the screen; blue, red, yellow, green. Each time the word appears, it is presented in a different colour; blue, red, yellow or green. Timed task over 60 s, user required to select correct colour of word. Scored by number of correct answers. No marks lost for incorrect answers.

**Table 6 behavsci-13-00717-t006:** Baseline demographic and clinical characteristics.

Demographics and Characteristics	*n* (%)	Mean ± SD	Median (IQR)
Sex Female Male	14 (70) 6 (30)		
Age (years)		47.3 ± 10.9	49 (41–53)
Symptom duration (years)		15.6 ± 7.7	14.5 (10–20)
Marital status Married Single Divorced Co-habiting Civil partnership	10 (50) 6 (30) 1 (5) 2 (10) 1 (5)		
Employment status Employed full-time Employed part-time Self-employed Unemployed (looking for work) Unemployed (not looking for work) Sick leave Retired	4 (20) 1 (5) 2 (10) 1 (5) 7 (35) 1 (5) 4 (20)		
Education level Some secondary school Completed secondary school Completed further education (sixth form) Higher education	1 (5) 2 (10) 1 (5) 16 (80)		
Ethnicity Asian or Asian British Black British White British	5 (25) 1 (5) 14 (70)		
Measurements Height (cm) Weight (kg) BMI ((kg/m^2^) Systolic blood pressure (mmHg) Diastolic blood pressure (mmHg) Heart rate Oxygen saturations (%)		166 ± 10.1 87.9 ± 19.1 31.5 ± 5.9 136 ± 20.9 86 ± 10.9 79 ± 12.0 98 ± 1.0	

**Table 7 behavsci-13-00717-t007:** Mean change pre- and post-trial intervention (Week 6), presented with 95% Confidence Intervals and effect size of change.

Outcome Measure	Mean Improvement (95% CI)	Cohen’s *d* Effect Size
Participant-reported outcome measures		
Revised Fibromyalgia Impact Questionnaire FIQR Stiffness FIQR Dyscognition	24.44 (15.27 to 33.60) * 3.11 (2.05 to 4.16) 2.74 (1.48 to 3.99)	1.49 * 1.59 1.38
Brief Pain Inventory—Short Form BPI Pain Intensity BPI Pain Interference	3.01 (2.38 to 3.64) 2.35 (1.31 to 3.39)	2.37 1.43
Fibromyalgia Severity Score	8.68 (5.61 to 11.76)	1.95
Fatigue Severity Scale	0.67 (0.04 to 1.39)	0.68
Jenkins Sleep Questionnaire	5.68 (2.84 to 8.53)	1.27
Hospital Anxiety and Depression Score HADS-A HADS-D	3.47 (2.02 to 4.93) 4.21 (2.65 to 5.77)	0.83 1.23
Performance-based outcome measures		
Tender Point Examination Fibromyalgia Intensity Score Total Pressure tolerated (kg/cm^2^)	1.08 (−0.03 to 2.19) 0.57 (0.16 to 0.99)	0.52 0.49
Stroop Test Total Score Accuracy (%)	4.11 (0.61 to 7.60) 0.70 (−7.21 to 8.61)	0.24 0.01

******* *p ≤* 0.001.

**Table 8 behavsci-13-00717-t008:** Mean changes at Week 24, presented with 95% Confidence Intervals and effect size of change.

Outcome Measure	Mean Improvement (95% CI)	Cohen’s *d* Effect Size
Revised Fibromyalgia Impact Questionnaire Week 6/Week 24 Baseline/Week 24	−10.41 (8.98 to 11.85) * 14.02 (12.55 to 15.49) **	0.57 0.94
Patient Global Impression of Change Week 6/Week 24	1.68 (1.47 to 1.9)	0.94

* *p* = 0.23, ** *p* = 0.001.

**Table 9 behavsci-13-00717-t009:** Overview of drug classes and dose changes during course of trial intervention.

DRUG CLASS	Reduced (or Stopped)	Static	Increased
Paracetamol, *n* = 6	1 (2)	2	1
Anti-inflammatories, *n* = 4	1 (1)	1	1
Opioids, *n* = 17	6 (3)	6	2
Tricyclic antidepressants (TCAs), *n* = 11	1 (1)	8	1
SSRIs/SNRIs, *n* = 11	0 (2)	8	1
Anticonvulsants, *n* = 11	1 (0)	9	1
Anxiolytics, *n* = 3	0	3	0
Sleeping tablets, *n* = 3	0	3	0
Beta blockers, *n* = 2	0	2	0
Migraine prophylaxis and treatment, *n* = 3	0	3	0
Antipsychotic, *n* = 1	0	0	1

‘*n*’ denotes number of drugs.

## Data Availability

The datasets used and/or analysed during the current study are available from the corresponding author on reasonable request.

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
