# Peer review of "Whole-Body Photobiomodulation Therapy for Fibromyalgia: A Feasibility Trial"

_behavsci, 2023, doi:10.3390/bs13090717_

Round 1

Reviewer 1 Report

The manuscript, however very interesting, needs a thorough tidying up and clearing of redundant details that interfere with the essential content.

1. Descriptions of the eligibility criteria, questionnaires and other research tools should be included in the chapter "material and methods" and should not be repeated in other chapters.

2. The "results" should not contain descriptions of the tested material, and the discussion should not repeat the results or research procedures. The discussion should also be organized according to a clear keynote, which will allow for a concise summary of own research achievements and in the light of the literature on the subject.

3. Objective vaguely formulated and it is not known whether it has been achieved, as it is difficult to define the meaning of the phrase "reflective thematic analysis - separate and inductive".

4. 4. Conclusions not specific and do not refer to own results. The authors do not even in this part attempt to organize the manuscript in a sensitive way.

Author Response

Thank you for the time given to your review and your important points made.

The manuscript, however very interesting, needs a thorough tidying up and clearing of redundant details that interfere with the essential content.
Author response: Thank you for your positive feedback. The manuscript has been considered for redundant content and updated.

Descriptions of the eligibility criteria, questionnaires and other research tools should be included in the chapter "material and methods" and should not be repeated in other chapters.
Author response: Thank you for this comment this information has been updated.

The "results" should not contain descriptions of the tested material, and the discussion should not repeat the results or research procedures. The discussion should also be organized according to a clear keynote, which will allow for a concise summary of own research achievements and in the light of the literature on the subject.
Author response: Thank you for these important observations we have removed any methods from the results section and results from the discussion. We have made sure the focus of the discussion is clear.

Objective vaguely formulated and it is not known whether it has been achieved, as it is difficult to define the meaning of the phrase "reflective thematic analysis - separate and inductive".
Author response: Thank you for this observation we have removed the term separate and inductive so the reader is not confused by this deviation from the original term.

Conclusions not specific and do not refer to own results. The authors do not even in this part attempt to organize the manuscript in a sensitive way.
Author response: We have made the conclusion more specific.

Reviewer 2 Report

-          Small Patient Group (49 patients screened, 21 paticipants, 19 Completions) –  

-          Descriptive Analysis no prospective double blinded – therefore Placebo-effect?

-          NO Follow-up bloodwork

-          FIQR increases in Week 24  found no viable explanation why ?

-          No reduction in antidepressants

Author Response

Thank you for your comments and observations. We have responded to your comments below:

Small Patient Group (49 patients screened, 21 paticipants, 19 Completions) – Author response: Thank you the sample size was not powered for a full trial due to the nature of the type of trail being a pilot study. We have added this as a limitation.

Descriptive Analysis no prospective double blinded – therefore Placebo-effect?Author response: This has been added as a limitation.

NO Follow-up bloodwork
Author response: This has been added as a limitation.

FIQR increases in Week 24  found no viable explanation why ?
Author response: This has now been considered in the discussion.

No reduction in antidepressants.
Author response: No changes made. Although no SSRIs/SNRIs were reduced - 2 participants stopped them (in brackets). 

Reviewer 3 Report

This study is a preliminary! One. There are a lot of limitations like the number of participants, high dropout rate , selection bias of high intensity FM patients and more . The most important limitation of this study in addition to the limitations stated by the authors,is lack of placebo treatment which reduces dramatically the scientific value of this trial.I am not convinced about the soundness of the patients' diagnosis as well. It is well known the value of controlling especially in FM patients.

Author Response

Thank you for your comments and observations. We have included these into the manuscript.

This study is a preliminary! One.
Author response: it is a feasibility study with feasibility aims. This has not yet been established the subsequent weaknesses are part of feasibility work. It would be unethical to perform a full scale multi-centre trial prematurely.

There are a lot of limitations like the number of participants, high dropout rate , selection bias of high intensity FM patients and more . The most important limitation of this study in addition to the limitations stated by the authors, is lack of placebo treatment which reduces dramatically the scientific value of this trial. I am not convinced about the soundness of the patients' diagnosis as well. It is well known the value of controlling especially in FM patients.
Author response: Thank you for identifying these limitations we have ensure they are all acknowledged in the manuscript.

Round 2

Reviewer 3 Report

Accept